# Electronic Health Information Systems to Improve Disease Diagnosis and Management at Point-of-Care in Low and Middle Income Countries: A Narrative Review

**DOI:** 10.3390/diagnostics10050327

**Published:** 2020-05-20

**Authors:** Thokozani Khubone, Boikhutso Tlou, Tivani Phosa Mashamba-Thompson

**Affiliations:** 1Department of Public Health Medicine, School of Nursing and Public Health, University of KwaZulu-Natal, Durban 4041, South Africa; Tlou@ukzn.ac.za (B.T.); Mashamba-Thompson@ukzn.ac.za (T.P.M.-T.); 2Department of Public Health, Faculaty of Health Sciences, University of Limpopo, Polokwane 0727, South Africa

**Keywords:** electronic health information system, diagnosis, treatment, point-of-care, low and middle income countries

## Abstract

The purpose of an electronic health information system (EHIS) is to support health care workers in providing health care services to an individual client and to enable data exchange among service providers. The demand to explore the use of EHIS for diagnosis and management of communicable and non-communicable diseases has increased dramatically due to the volume of patient data and the need to retain patients in care. In addition, the advent of Coronavirus disease 2019 (COVID-19) pandemic in high disease burdened low and middle income countries (LMICs) has increased the need for robust EHIS to enable efficient surveillance of the pandemic. EHIS has potential to enable efficient delivery of disease diagnostics services at point-of-care (POC) and reduce medical errors. This review provides an overview of literature on EHIS’s with a focus on describing the key components of EHIS and presenting evidence on enablers and barriers to implementation of EHISs in LMICs. With guidance from the presented evidence, we proposed EHIS key stakeholders’ roles and responsibilities to ensure efficient utility of EHIS for disease diagnosis and management at POC in LMICs.

## 1. Introduction

The health sector is lagging behind in the era of information and technology (IT). The main purpose for use of IT in the health sector include the following: extending geographic access to health care; enhancing client communication with the health provider; improving disease diagnosis and treatment; improved data quality management; and to avoid fraud and abuse of client’s confidentiality [1,2,3]. The introduction of digitization has revealed the possibilities and costs benefits to health care management. IT systems such as electronic health information systems (EHIS) have been shown to be a useful tool for improving disease diagnosis and treatment at point of care (POC), globally [4,5,6]. EHIS is the digital version of a patients’ paper chart, which has capacity to store health data such as test results and treatments. It is also designed to enable real-time, patient-centered records that make information available instantly and securely to the authorized users [7]. The term EHIS is used interchangeably with electronic health records (EHRs), eHealth and electronic medical records (EMRs). EHIS are a vital part of health IT built to go beyond standard clinical data collected in a providers’ office and can be inclusive of a broader view of a patient care [8].

An efficient functioning EHIS requires the use of digital health systems such as three interlinked electronic register (Tier.Net), which has an ability to facilitate information exchange between software [9]. Tier.Net is used by healthcare facilities to enable electronic collection, storage, management and sharing of patient’s electronic health or medical records for the purpose of patient care, research and quality management [10]. Countries are currently battling with a global pandemic caused by the outbreak of SARS COV-2, a virus that causes Coronavirus disease 2019 (COVID-19). The advent of COVID-19 in high disease burdened low and middle income countries (LMICs) such as South Africa has increased the need for robust EHIS to enable efficient surveillance of the pandemic [11]. The main objective of this review is to presents an overview of literature on the characteristics of EHIS and implementation of EHISs for improving disease diagnosis and treatment at point-of-care in the LMICs. We search for literature from the following databases: PubMed and Google Scholar and included relevant literature from LMICs.

## 2. Characteristic of Electronic Health Information Systems

An efficiently functioning EHIS is key to health service delivery as it promises a number of substantial benefits, including improving the quality of healthcare service delivery, decreased healthcare costs as well as reduce serious unintended consequences [12]. A poorly implemented EHR system can lead to EHR-related errors that jeopardize the integrity of the information in the EHR, leading to errors that endanger patient safety as well as compromise the quality healthcare services [12]. The following key components are required for an efficient functioning EHIS: patient management component; activity component; clinical component; pharmacy component; laboratory component; radiology information system; and billing system (Figure 1) [13]. Table 1 provides a description on the functions of EHR components within the electronic health system and patient care.

## 3. Opportunities Presented by EHIS in the LMICs

Evidence on EHIS in developing countries revealed the following eHealth attributes: tracking of patients who were initiated on treatment; monitoring of adherence to care and early detection of potential loss to follow up; minimize the time it takes to communicate data between different levels; reduction of errors especially the laboratory data; linkage to bar code for unique identification and laboratory samples and the prescription of medication [18]. In Mozambique, a robust electronic patient management system facilitated a facility-level reporting of required indicators, improved ability to identify patients lost to follow-up; and support facility and patient management for HIV care [19]. An implementation study aimed at implementing an integrated pharmaceutical management information system for antiretroviral treatment (ART) and other medicines in Namibia showed the system’s reliability in managing ART patients, monitoring ART adherence and HIV drug resistance early warning indicators [20].

## 4. Enablers of EHIS Implementation in the LMICs

Enables of EHIS implementation in the LMICs are aligned with leadership abilities, sound policy decision and financial support with the goals of purchasing IT, connectivity and capacity building [21]. Enablers for EHIS in LMICs includes: legislation, financial investment; staff training, political leadership; acceptability of technology; performance expectancy; and social influence among professionals [22,23,24].

### 4.1. Financial Investment

Many LMICs are supporting financial investment to help scaling up of EHIS. A study from China recommended that in order to achieve the national childhood immunization information management system objectives for 2010, the funding for system-building should be increased [22]. A three-country qualitative study was conducted in southern Africa on the sustainability of health information systems which revealed; more government commitment in funding EHIS such as printer ink, IT infrastructure, recruitment of personnel and running costs [23]. In Ghana, cooperation between the vendors and management was demonstrated [25]. This successful cooperation translated into regularly provision of feedback and sucessful system maintenance [25]. This has helped the facility in alleviating the common challenge faced by most Information Communication and Technology (ICT) implementers in LIMCs [25].

### 4.2. Legislation

South Africa National Health Act of 2003 is a good example of a legislation, policy, norms and standards defining the role of national, provincial and local governments in terms of EHIS implementation in LMICs [26]. South Africa has advocated the scale up of digital health technologies to improve access to health care and for health systems strengtherning through systems such as Tier.net and District Health Information Software (DHIS and patient registration systems [27]. The delivery of EHIS or eHealth in South Africa’s public sector facilities is the responsibility of the provincial departments of health, while policy development resides with the National Department of Health (NDoH). In terms of Section 74 of the National Health Act, the NDoH is also responsible for facilitation and coordination of health information. 

### 4.3. Staff Training

There is growing evidence on the value of well-trained health informatics workforce in LMICs [24]. Studies conducted in Botswana and Uganda showed the on-the-job training and mentorship as a major enabler for EHIS in LMICs [28,29]. This were shown to be an effective approach for strengthening monitoring and evaluation capacity and ensuring data quality within a national health system [28]. It was demonstrated that on-the-job training can also improves performance through timely and increased reporting of key health indicators [29].

### 4.4. Political Leadership

Effective leadership can positively contribute to the successful adoption of new EHIS in any organization [30]. In Ethiopia, the role of ICT towards universal health coverage prompted academic and political spheres to make ICT on the agenda especially for disease diagnosis and treatment in the LMICs [31]. The Rwandan government has also shown commitment to telemedicine, through their strategic choice of using low-cost and less complex technologies, and strategic partnerships with educational and technology companies to help in the implementation of telemedicine [32].

### 4.5. Acceptability of Technology

Research has shown that factors such as English language proficiency level, computer literacy and EMR literacy level and education level can influence the level of use of EHIS [33]. Liu and others revealed that the usage of EHIS by health workers in LMICs can be influenced by the level of system simplicity and user friendliness [34]. An economical mobile health application to improve communication between healthcare workers was introduced in KwaZulu-Natal, South Africa using an iterative design process [35]. This application was received positive feedback from healthcare workers due to its ability to improve team spirit between community and clinic based staff [35].

## 5. Barriers and Challenges to Implemention of EHIS in the LMICs

There are various factors impeding the successful implementation and scale up of EHIS in LMICs. These include the following: complexity of the intervention and lack of technical consensus; limited human resource, poor leadership, insufficient finances, staff resistance, lack of management, low organizational capability; misapplication of proven diffusion techniques; non engagement of both local users and inadequate use of research findings when implementing [36].

### 5.1. Complexity of the Intervention and Lack of Technical Consensus

The complexity of the EHIS which and lack of consultation as key barriers on the implementation in LMICs [36]. Designing an organizational EHIS with a complex design is a serious threat of the implementation in LMICs [37]. In Rwanda, the interfaces between the existing and new EHIS are the inhibitors to the implementation [38]. There are instances of patient information that are captured into the computer; but challenged with bandwidth requirements in health facilities [39].

### 5.2. Limited Human Resource

The main barriers in implementing EHIS on the LMICs relate to lack of capacity: human, leadership and management [36]. Human resource capacity is the main barrier not only in terms of the supply but also in terms of the ability to perform the task. The exodus of skilled cadres to the well-paying non-government organizations are the contributing factors to human resource capacity [40].

### 5.3. Lack of Management

Ineffective coordination, poor management and lack of supervision for EHIS are the main challenges in the LMICs [41,42]. Management capacity and the ability to use data were reported as the root causes in facilities with inadequate human resource, computers and data capturing skills [43]. Late submission of health data and absence of feedback from the supervisors are the key barriers to EHIS implementation in LMICs [44].

### 5.4. Lack of Funds

EHIS implementation is costly as there is hardware, software, maintenance, training and human resource investment making implementation unaffordable to many LMICs [45]. Cost is the main constraint to adoption and implementation of EHIS in LMICs [46]. Running costs and political will are the prerequisite for sustaining EHIS [40]. Unreliable electricity supply, shortage of IT equipment, poor connectivity and safe accommodation for the equipment are the restraining elements to the successful implementation of EHIS [45].

### 5.5. Inadequate Health Systems Capacity

Poor public healthcare system with ever changing policies are a hindrances to the successful implementation of the EHIS in LMICs [21]. Leon and others used a framework for assessing the health system challenges to scaling up m-Health in South Africa and revealed a weak ICT environment and limited implementation capacity within the health system [47]. Katuu explored the barriers in improving South African public health sector through eHealth strategy particularly by integrating electronic document and records management system. Inequality, historical red tape and curative structure are the main barriers [48].

### 5.6. Poor Application of Proven Diffusion Techniques

In Asia, incapacitated human resources and shortage of IT skills were identified as inhibiting factors to EHIS implementation [49]. In Iran, lack of users’ knowledge about system and working with it were the barriers identified [50]. In most of the LMICs; the need for a trained workforce in health informatics is great [51]. There are instances where computer illiterate and low morale to use the system are affecting the implementation [36,52]. Some of the challenges include related to EHIS software, cost drivers, interoperability, connectivity in rural set up and data quality [40].

### 5.7. Staff Resistance

A study conducted in South Africa, demonstrated difficulties with implementing a dual EHIS as a result of clinicians’ resistance to using the EHIS and feel more comfortable using paper based system [52]. In Iran, the negative staff attitudes of system developers and lack of acceptability are the main barriers to successful implementation of hospital-based EHIS [50]. Although South Africa EHIS catered for all required information, the hospital officials show poor due to the attitude and resistance to using EHIS for patient treatment and prescriptions [53]. An assessment was conducted by Khasi EHIS state of readiness for rural South African areas, which revealed that the resistance to change and negative perceptions were two key causes for not accepting the intervention. Any new EHIS intervention must address them in order to succeed [54].

### 5.8. Compromised Data Quality

Studies revealed incompleteness of TB data across multiple information systems in South Africa. Variances between 12% and 38% of the missed cases due to poor recording from the source documents (either patient records or laboratory records) were demonstrated [49,50]. Data collected and reported in the public health system across three large, high HIV-prevalence districts was neither complete nor accurate enough to guide patient tracking as part of prevention of mother to child transmission (PMTCT) care [51]. 

## 6. Discussion

This review has provided us with a great platform to depict opportunities of EHIS implementation in LMICs. It has also enabled us to identify and classify barriers and challenges implementation of EHIS that must be addressed pre-implementation to ensure the success. Key to the success of EHIS is the leader’s willingness to play a leading role in adopting data demand and supply principles for decision making. The presented literature reveals the need for well-defined roles of EHIS stakeholders to ensure successful implementation and utility. Here, we proposed key stakeholders roles and responsibilities in the implementation of EHIS for disease diagnosis and management at point-of-care (POC) in LMICs (Figure 2). In the proposed key stakeholders’ roles and responsibilities we emphesise on that the information culture should be cascaded through different hierarchy levels of an organization. In the absence of the such culture there is likely to be poor adoption, poor data quality and utilization [55].

## 7. Conclusions

The advent of EHIS has revolutionize patient care through improving both disease diagnosis and treatment at POC. However, its use in LMICs is still limited, despite the high disease burden in these settings. EHIS implementation need to be one of the global health priorities to help respond to community’s health needs, particularly during the current Covid-19 pandemic. Successful implementation of EHIS requires commitment from health leaders to play a strategic role in terms of the policy directive, resource mobilization and evidence-based decision-making. To help optimize the implementation and use of EHIS in LMICs, we have proposed roles and responsibilities of stakeholders to ensure efficient and sustainable implementation of EHIS. A systematic approach for stakeholder engagement would be crucial to ensuring successful operationalization of the proposed roles and responsibilities.

## Figures and Tables

**Figure 1 diagnostics-10-00327-f001:**
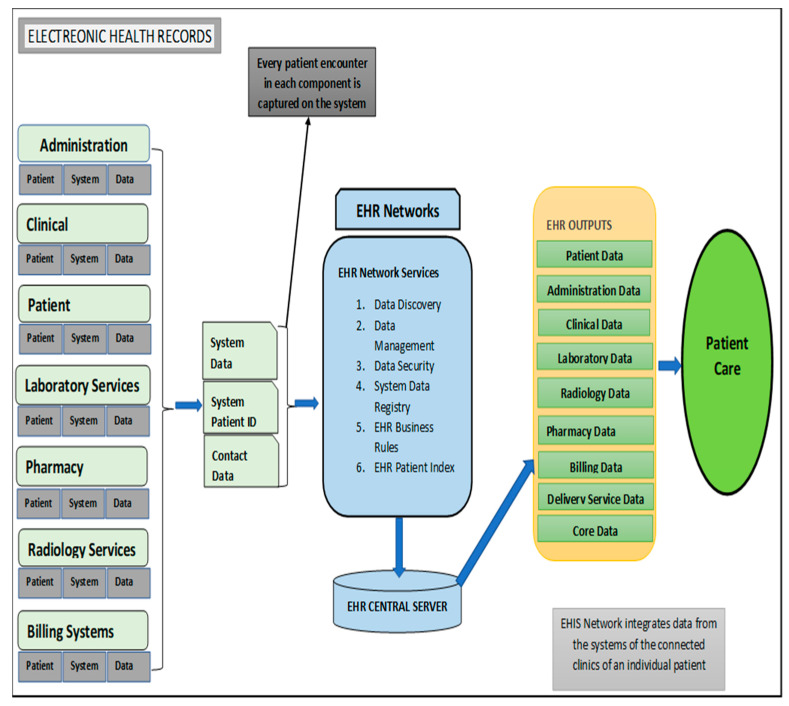
Overview of electronic health information system components (adapted from the National Institute of Health National Center for Reasearch Resources; 2006 [13]).

**Figure 2 diagnostics-10-00327-f002:**
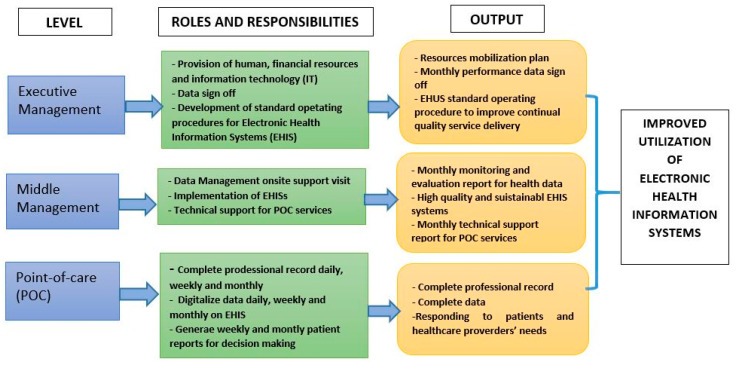
Proposed roles and responsibilities of stakeholders to ensure efficient utility of electronic health information systems for disease diagnosis and treatment at point-of-care in low and middle income counties.

**Table 1 diagnostics-10-00327-t001:** Description of electronic health record (EHR) components, purposes within the electronic health system and patient care by National Institutes of Health National Center for Research Resources.

EHIS Component	Function	Benefit to Patient Care
Patient Management EHIS	Patient registration, admission, transfer and discharge (ADT) functionality. Patient registration includes key patient information such as demographics, insurance information and contact information [14]	Populations and their needs are analyzed at a point of care to determine the services to be rendered to them [15]
Activity EHIS	Flow processed from when a client is entering the point of service till data is digitized on the system [15,16]	Traceability of health data
Clinical EHIS	Habitation of multiple sub-components, e.g., computerized provide order entry (CPOE), electronic documentation, nursing component [14]	Electronic clinical documentation systems enhance the value of EHRs by providing electronic capture of clinical notes; patient assessments; and clinical reports, such as medication administration records (MAR) [13]
Pharmacy EHIS	Islands of automation, such as pharmacy robots for filling prescriptions or payer formularies, that typically are not integrated with EHRs [13]	Improve efficiency of pharmacy services
Laboratory EHIS	Consists of two subcomponents: capturing results from lab machines; and integration with orders, billing and lab machines. The lab component may either be integrated with the EHR or exist as a standalone product [14,17]	Improve efficiency of pathology laboratory services
Radiology Information System and Picture Archiving & Communications System (PACS)	Manages patient workflow, ordering process and results [14]	Enables improved service delivery
The billing system (hospital and professional billing)	Captures all charges generated in the process of taking care of patients. These charges generate claims, which are subsequently submitted to insurance companies, tracked and completed [14]	Tracking of patient data and quality assurance

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
