# Peer review of "Electronic Health Information Systems to Improve Disease Diagnosis and Management at Point-of-Care in Low and Middle Income Countries: A Narrative Review"

_diagnostics, 2020, doi:10.3390/diagnostics10050327_

Round 1

Reviewer 1 Report

The manuscript describes the essential aspects of the EHIS in an organized manner covering all the important aspects of it and describing the key details and insights. It is a very nice manuscript that covers the mentioned topic perfectly.

Just a few minor comments that will make it more appealing to the readers.

  1. EHIS is a main component of all the emerging smart and POC devices in healthcare. The authors should mention some lines on this aspect in the introduction. Some relevant references would be Trends in Biotechnology 33(11), 692-705, 2015; Diagnostics, 4(3), 104-128; https://link.springer.com/book/10.1007/978-3-030-11416-9
  2. The authors can mention the cost-effectiveness provided by the use of EHIS to the healthcare and reimbursement agencies, which are increasingly getting adapted to smart solutions.
  3. A recent example is the use of EHIS internationally for COVID-19 that is interfaced to smart technologies. It would be interesting if authors can mention some specific utility scenarios where the advances in EHIS have led to increased benefits. This could be in the last section before conclusions.

Author Response

We much appreciate the reviewer's evaluation of our work. Please see our response in the attachment.

Reviewer 2 Report

Thank you for the opportunity to review this article.

This review article sets out to provide an overview of literature of EHIS implementations and the barriers and facilitators to such implementations. Based on their review findings, the authors have put forth a framework for enhanced utility of point-of-care EHIS in LMIC.

Overall, the review would benefit from restructuring of content into 1) introduction/background and objectives, 2) methods, 3) results and 4) discussion. It is unclear to me if some of the sections are findings from the review that the authors did, or are an overall description that belong in the background section. The authors seem to migrate from one framework to another, and this is very confusing to follow. Throughout the review, there are repetitive themes that need to be grouped thematically, and perhaps based on a purposefully selected theoretical framework. I strongly advice the use of uniform terminology throughout the review to refer to themes and sub-themes, in line with widely cited literature such as the WHO eHealth strategy toolkit and WHO guideline recommendations on digital health interventions, among others.

Detailed comments below: 

Line 15: there is a missing word here. The expansion of LMICs is Low and middle income countries.

Line 20: This sentence “With guidance from the presented literature; we proposed a framework for improving the utility of EHIS for disease diagnosis and management at POC in LMICs” should read as “With guidance from the presented literature; we propose a framework for improving the utility of EHIS for disease diagnosis and management at POC in LMICs”.

Line 32: “It is also designed to enable a real-time, patient-centred records that make information available instantly and securely to the authorized users” should read as “It is also designed to enable real-time, patient-centred records that make information available instantly and securely to the authorized users”.

Line 39: Could you state briefly what the ‘three interlinked electronic register’ is? I see that you have added an appropriate reference, but would be helpful to add a few words in the text.

Line 45: You can drop the extra preposition in the sentence. “….treatment at point-of-care in LMICs”.

Line 47: Is there a rationale for a detailed description of the concept of routine health information systems based on your review? Is this not already described reasonably well in the literature?

Line 53: How come other keywords such as Electronic Medical Records/ Electronic Health Records were not included? As far as I know, the MeSH terms for a PubMed search are 'Electronic Medical Records' or 'Electronic Health Records'.

Line 54: Could you state how many studies, journal articles, gray literature and documents you found based on your search criteria? What was the methodology behind screening of the articles? Were there any inclusion (or exclusion criteria) for articles that were finally included your analysis? Some clarifications of the exact review methodology are essential.

Lines 66-70: please check for typos, sentence grammar and tense.

Figure 1 and table 1: are these based on a synthesis of your review findings? How did you choose the themes to be presented in table 1?

Line 75: the main purposes you have mentioned here – are they thematic extractions from the literature you reviewed? Would it be possible to add citations against each of the purposes mentioned?

The section on ‘Use of Information and Technology in the Health Sector’ is a bit confusing. The narrative presents some case studies, but also includes implementation considerations such as enablers and barriers of EHIS. Considering you have a separate section on enablers and barriers, I suggest you remove these from this section. For example, line 113 describes a clear enabler of EHISs.

Line 175-183: this reads more like an introduction than actual findings from the review. I suggest you skip this section or move it to the overall background/introduction. See my comment about the overall structure.

Figure 2: is missing a citation. Is this an adaptation of the PRISM framework proposed by Aqil et al? The reference is attributed to another author, so I am unsure. Did the authors of this review make this adaptation in figure 2? Why did you choose to present this framework in particular?

Lines 204-245: the themes you present under the behavioral, organizational, and technical factors – these are also enablers or barriers of EHISs in general, are they not? Why are they presented here in a separate sub-section?  

Line 246: this sub-section seems to be out of place. ‘Impact on patient management and loss to follow up’ is also a ‘use of Information and technology in the health sector’. I am not sure if there is a reason for keeping it as a separate sub-section.

Line 262: Are these the themes that emerged from your review? Are these the enablers at the health system level? One of the important enablers to the use of point-of-care systems and for data use is the existence of legislation and regulatory frameworks. The WHO eHealth strategy toolkit also identifies having a long-term strategy as an important enabling factor. Did these not emerge in your review findings?

The enablers seem to be a mix of more system level factors like political leadership and downstream enablers like social influence among professionals, which affects health workers and end users. What about other factors that affect acceptability and feasibility of digital health interventions for health workers? The review accompanying the WHO Guideline Recommendation for Digital Health Interventions identified many more of these factors.

Line 312: are these participants of the primary study you are referring to?

Line 322: same as the comment on line 312.

Line 335: this is a bit redundant, since political leadership is already mentioned as an enabler. The lack thereof would naturally be a barrier.

Line 390: since the development of a framework is part of the objectives, I believe this should be presented as a result and not as part of the discussion.

Figure 3: I am struggling to see the direct link between the findings presented in your review thus far and the framework that the authors have created (based on the review findings?). For example, ‘data sign-off’ is mentioned as one of the roles and responsibilities of ‘executive management’. Yet, as far as I can see, this is not presented as a barrier/enabler of EHISs. ‘Standard operating procedures’ is another such theme mentioned in the framework for the first time. On the other hand, training is rightfully mentioned as an important enabling factor, but not included in the framework in figure 3.

The framework itself is perhaps missing a few links between the different layers. For example, ‘high quality and sustainable EHR system’ is an output of several factors, and not just an output of 'middle management'. By 'improved utilization of electronic health information systems', do the authors mean point-of-care use of EHIS for data entry or data use for decision-making?

I would urge the authors to reconsider the use of the word ‘framework’ in this context. As you rightly describe in line 405, figure 3 only describes the roles and responsibilities of some stakeholders. If this should be presented as a framework, then it is missing a few more steps/layers as well as some arrows. For instance, ‘creating an data or information culture’, that you have mentioned later on should be a part of figure 3, if this is in fact a framework.

Line 419: You mention that “EHIS intervention will still fail if the government and policy makers are not playing their strategic role in terms of the policy directive, resource mobilization and increase data demand and supply for evidence based decision making in the LMICs”. In that case, this component should be included in the framework presented in figure 3.

A discussion section should be added, where the authors reflect on the findings of their review in light of other published literature, relevant reviews and frameworks. This section should typically include a description of strengths and weaknesses of the present study.  

References: There are some duplicate references (reference number 96 and 97, for example). Please go through your reference list and the in-text citations carefully to remove duplicate references. The bibliography also needs formatting. Some references are missing journal information, for instance. 

Author Response

(The authors gave the same response as above.)

Round 2

Reviewer 2 Report

Thank you for the opportunity to review the revised manuscript. 

The authors have addressed all the concerns I had raised in my review, and the changes are reflected in the revised manuscript. 

There are some minor typos and English language corrections, which I believe can be rectified with a careful read-through of the manuscript. 

I do not have any more comments.